# Ruminating Word Representations with Random Noise Masking

## Abstract

We introduce a training method for better word representation and performance, which we call **GraVeR** (**Gra**dual **Ve**ctor **R**umination). The method is to gradually and iteratively add random noises and bias to word embeddings after training a model, and re-train the model from scratch but initialize with the noised word embeddings. Through the re-training process, some noises can be compensated and other noises can be utilized to learn better representations. As a result, we can get word representations further fine-tuned and specialized in the task. On six text classification tasks, our method improves model performances with a large gap. When GraVeR is combined with other regularization techniques, it shows further improvements. Lastly, we investigate the usefulness of GraVeR[1].

## 1 Introduction

Most machine learning methodologies can be formulated to get computational representations from real-life objects (e.g., images, languages, and sounds) and then get high-level representations using model architectures. Therefore, there have been two main approaches to improve model performances: (1) starting with better representations (Melamud et al., 2016; Peters et al., 2018), and (2) building more sophisticated architectures that can extract important features and generate higher-level representations (Vaswani et al., 2017; Conneau et al., 2017).

For better initial representations, many NLP researchers have used pretrained word vectors trained on substantially large corpus through unsupervised algorithms, such as word2vec (Mikolov et al., 2013a), GloVe (Pennington et al., 2014), and fastText (Bojanowski et al., 2016). The pretrained word vectors represent the general meaning of words and increase the model performances on most NLP tasks (Turian et al., 2010). In addition to the algorithms, word vector post-processing research (Faruqui et al., 2015; Vulić et al., 2017; Mrkšić et al., 2017; Jo & Choi, 2018) have attempted to enrich the pretrained representations using external resources. They simply modified the values of the vector representations in some way, and showed improved performance. It implies that we can get further improvement through better initial representations.

When training NLP models, we first initialize word representations with pretrained word vectors and then update both the model parameters and the word representations. However, in the training process, the model performance can be limited due to the initial word vectors. For example, the pretrained word representations have general meanings of words, but, in some tasks, the words might not be used for the general meaning. Although the gap between meanings can be learned through the training process, it could fail. Since the pretrained representations are trained from a huge dataset, and their objective functions are based on language modeling, the word vectors are naturally biased to general and frequently used meaning. Besides, the word vectors are updated through gradient descent algorithms, so the values are changed slightly. The word vectors are thus easy to converge on local minima.

Therefore, our method starts with an idea–using the word representations fine-tuned by a training process as pretrained word vectors in the next re-training process. Then, word vectors can be trained to learn more appropriate representations to the task. However, the model must be overfitted, and then the word representation would be stuck in local minima. Thus, we add random noise and bias to the word representations before the re-training processes, in order to prevent the model from overfitting and take the word representations far from the local minima.

---

[1] http://github.com/Sweetblueday/GraVeR

In this paper, we propose a simple training framework to find better representations by adding random noise and bias on the word vectors during iterative training processes, which we call **GraVeR** (**Gra**dual **Ve**ctor **R**umination). We expect that the model makes good uses of the re-training processes with noises, both for learning better representation and for model regularization.

## 2 RELATED WORKS

The representations fine-tuned by GraVeR can be considered as pretrained representations from the previous training process. Also, GraVeR utilizes word-level noises, which are used for model regularization.

### 2.1 PRETRAINED REPRESENTATIONS

**Pretrained Embedding Vector** is also called pretrained word representation. According to the distributional representation hypothesis (Mikolov et al., 2013b), pretrained embedding vectors are composed of pairs of (token, n-dimensional float vector). Unsupervised algorithms (e.g., word2vec (Mikolov et al., 2013a), GloVe (Pennington et al., 2014), fastText (Bojanowski et al., 2016)) learn the word vectors on substantial corpora to represent general meanings of words. The pretrained embedding vectors are widely used to initialize the word vectors in models.

**Pretrained Embedding Model** is suggested to get a deep representation of each word in the context. Previous research McCann et al. (2017); Peters et al. (2018); Devlin et al. (2018) trained deep architecture models and then utilized the model weights to represent words by using the outputs of the models. Although recent advanced pretrained representations (Peters et al., 2018; Devlin et al., 2018) show good performances, we take the pretrained embedding vector approach because (1) re-training processes in the pretrained embedding models are very expensive and (2) we use word-level noises whereas the embedding models use token-level embeddings combined with position embeddings.

### 2.2 WORD-LEVEL NOISES

Adding noises to input data is an old idea (Plaut et al., 1986). However, a small number of studies were on word-level noises, since the noises on words can distort words' meaning.

**Word Dropping.** NLP tasks that utilize the text as a form of sentence and phrase have considered each word as features. However, too many features can lead models to be overfitted to the training data due to the curse of dimensionality. Therefore, the easiest way to reduce the number of features is to drop words in the sentence at random.

**Word Embedding Perturbation.** Miyato et al. (2016) tried to perturb word vectors and used them in the adversarial training framework for model regularization. Cheng et al. (2018) utilized the noises to build a robust machine translation model. Also, Zhang & Yang (2018) considered the perturbation as a data augmentation method. The previous works added the noises to all word embeddings. It can regularize the model weights, but they ignored the change of word representations and its re-usability.

On the other hand, our method gradually adds noises to word embeddings by controlling the amount of noise. Also, iterative training processes that re-use fine-tuned word representation as pretrained word vectors for the next training process can benefit from the noises and make better word representations.

### 2.3 REGULARIZATION TECHNIQUES

Some research explained that the normalization could be used for model regularization (van Laarhoven, 2017; Luo et al., 2018; Hoffer et al., 2018).

**Dropout** (Srivastava et al., 2014) is applied to neural network models, masking random neurons with 0. Dropout randomly and temporarily removes the neural activations during training, so the masked weights are not updated. As a result, the model is prevented from over-tuning on specific features, which involves regularization.

**Batch Normalization (BN)** (Ioffe & Szegedy, 2015) normalizes the features according to mini-batch statistics. Batch normalization enables the features to avoid covariate shift–the weight gradients are highly dependent on the previous layers' gradients. Besides, batch normalization speeds up

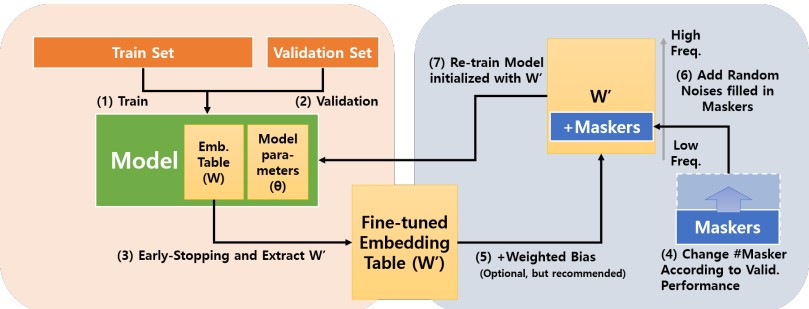

Figure 1: The flow of GraVeR. (1) We first train a classifier using the training set, (2-3) do early-stopping using the validation set, (3) extract fine-tuned word embeddings, (4) change the number of random maskers according to the validation performance, (5) add weighted bias to fine-tuned embedding $W'$, (6) add the maskers to a portion of $W'$, and lastly, (7) re-train the classifier with $W'$ from the very first step. We repeat the process until all the vocabularies are masked.

the training process by reshaping loss function.

**Layer Normalization (LN)** (Ba et al., 2016) also utilizes mini-batch statistics to normalize the features. The difference with batch normalization is that layer normalization normalizes the inputs across the features. The statistics are computed across each feature, which is the same for all the feature dimensions.

## 3 Proposed Method

### 3.1 Overall Process

The overall process of GraVeR is illustrated in Figure 1. GraVeR is applied to a conventional training framework, but it needs a meta-level approach that trains the model again. The iterative training process will also be denoted as a meta-epoch.

When a training process finishes, we extract the fine-tuned word embeddings ($W'$) and add bias ($W' - W$) weighted by 1/[MaskingRate]. The additive bias is optional but it takes the word embeddings far from the minima in which the fine-tuned embeddings converged. Next, we add maskers filled with random values to a portion of $W'$, and then re-train the model from scratch with the noised word embeddings. Observing the validation performance in the training process, we select the best-performing model and the fine-tuned embeddings. The additional details in GraVeR will be described. In short, GraVeR is a training framework that adds random noises (and bias) to the fine-tuned word embeddings and then repeats the training process with the word embeddings.

### 3.2 GraVeR Details

The details in GraVeR are based on the idea that moderate noises are required to change word vector distribution, but the noise scale should not be too large to distort the word vector distribution.

**Random Noises and Bias.** We have two hyperparameters related to the noise maskers; Masking Rate and Noise Scale. The masking rate denotes a portion of word vectors masked in every (re-)training process. We set the masking rate to 20% of the vocabulary size as default. The maskers are filled with random values in a range, which is denoted as noise scale. The default is sampled from a uniform distribution (-1,1) since word vectors in the range are widely used. Well-defined perturbation methods like Gaussian kernels (Vilnis & McCallum, 2014) can be an extension.

Next, we add a bias to the fine-tuned word embeddings ($W'$) in order to take $W'$ to the same embedding space with better initial values. Intuitively, since we initialize with the word embeddings fine-tuned in the previous training process, the embedding space in the next training process is the same as the space in the previous training process. And, the difference of fine-tuned embedding and initial embedding ($|W' - W|$) in the space means the representations learned by a training process, which improved the performance in the previous training process. So, we add the bias to $W'$, weighted by 1/[MaskingRate] where 0 <MaskingRate≤1 in order to give a large bias when a small number of word vectors are noised; 1/[MaskingRate] ×($W' - W$). When the validation performance

decreases, the bias is `[MaskingRate]` $\times (W - W')$ for small noise.

When selecting a portion of words to noise, we use word frequency driven from training data. Intuitively, if the representations of frequently used words are changed, which means the input data are changed a lot, the high-level representation such as sentence representation are largely affected.

**Gradualness.** While following the aforementioned process, we change the number of random maskers according to validation performance. If the re-trained model's validation performance increases, we move the maskers to the next frequently used words. Otherwise, we gradually increase the number of random maskers without moving them so that the maskers make noises on the words again in the next re-training process. As a result, GraVeR can process both words in the previous training process and words in the current training process. This gradualness lets the re-training processes benefit from newly generated noises again, which makes GraVeR dynamic.

In the end, overall algorithms of GraVeR are summarized as follows:

---

**Algorithm 1** Training Framework with GraVeR

---

$\mathbf{Acc}_{val} \leftarrow 0$, $\mathbf{MaxAcc}_{val} \leftarrow 0$
Train set (**Train**), Validation set (**Val**), A classifier ($M$), Word embeddings ($W$),
Word frequency information counted from training data
- Train $M$ with $W$
- Get trained $M'$ and fine-tuned $W'$
- Meta-level validation; $\mathbf{Acc}_{val} \leftarrow M'(\mathbf{Val}_{val}; W')$
**if** $\mathbf{MaxAcc}_{val} < \mathbf{Acc}$ **then**
    $\mathbf{MaxAcc}_{val} \leftarrow \mathbf{Acc}_{val}$
    **MaskedWords** $\leftarrow$ **NextFrequentlyUsedWords**
    $W \leftarrow W' + 1/\mathbf{MaskingRate} \times (W' - W)$
**else**
    **MaskedWords** $\leftarrow$ **MaskedWords** + **NextFrequentlyUsedWords**
    $W \leftarrow W - \mathbf{MaskingRate} \times (W' - W)$
**end if**
$W[\mathbf{MaskedWords}] \leftarrow W[\mathbf{MaskedWords}] + U(-1, 1)$
- Repeat the training process until all the words are masked

---

GraVeR can be applied to any conventional training frameworks, so the method is independent of model architectures in that most NLP models use word-level embeddings. The random noises might disturb the representation, but some of the noises which harm the performance are compensated during the re-training processes. In contrast, other noises are used to update the word vectors over the initial values. Besides, the additive bias makes the use of random noises stable.

By using this re-training with the noise process, we expect that the model is prevented from overfitting. Meanwhile, the model is incrementally fitted to the validation set through early-stopping in a training process and meta-level early-stopping in re-training processes. Therefore, the model keeps being fitted to the validation set with regularization, showing better performance on the test set since the model performance on the validation set is normally correlated to the performance on the test set.

## 4 EXPERIMENT

### 4.1 DATASETS

We prepare three topic classification datasets: **DBpedia ontology** (Lehmann et al., 2015), **YahooAnswer**[2] (Chang et al., 2008), **AGNews**. We also prepare two sentiment classification datasets: **Yelp reviews** (Zhang et al., 2015), **IMDB** (Maas et al., 2011). YahooAnswer dataset is used for two different tasks: classify upper-level categories and classify lower-level categories. The data statistics are presented in Appendix. We assign 15% of each train set to validation set, and each dataset has its own test set. The validation set is used for early-stopping both at every epoch and every meta-epoch. We use all words tokenized by space and all symbols using 300 dimensional embedding space.

---

[2]https://cogcomp.seas.upenn.edu/page/resource_view/89
Note that Chang et al. (2008) said the dataset has 20 top-level categories, but it has three duplicated top-level categories because of typos.

## 4.2 Classifier

We use TextCNN (Kim, 2014) classifiers. The model consists of 2 convolutional layers with 32 channels and 16 channels, respectively. We also adopt multiple sizes of kernels–2, 3, 4, and 5, followed by ReLU activation (Hahnloser et al., 2000) and max-pooling. The kernels are concatenated after every max-pooling layer. Finally, the features are classified into the classes using fully connected layer.

Although this model has few parameters (136K) compared with recent high-performance models like BERT (Devlin et al., 2018), we use this model to utilize multiple re-training processes. Also, we employ simple techniques for model improvements such as Dropout and Layer Normalization to get similar performances to recent models and justify the use of a simple model as a basic classifier. The vanilla classifier and the tuned classifier are denoted as TextCNN$_{base}$ and TextCNN$_{tune}$, respectively. Additional comparison with very deep models is described in Appendix.

We optimize the model using Adam (Kingma & Ba, 2014) with 1e-3 learning rate and early-stopping. If the validation accuracy does not increase over five epochs, we stop model training. Initial word embeddings are random if we do not mention explicitly.

## 4.3 Baseline Implementation

We cannot fully compare our method with **Word Embedding Perturbation** (Miyato et al., 2016) because we do not use adversarial training framework. Instead, random noises are added to all word embeddings, as other word embedding perturbation methods did (Cheng et al., 2018; Zhang & Yang, 2018).

In order to compare the effect of regularization, we implement five regularization (including normalization) methods. **Word dropping** is implemented in the pre-processing part, which removes random words in the text. We set the random probability $p$ as 0.1. **Dropout** (Srivastava et al., 2014) is added to the final fully connected layer with dropout probability 0.1, which performs the best in our experiments. **Batch Normalization** (Ioffe & Szegedy, 2015) is located between every convolutional layer and an activation function, as used in the original paper. **Layer Normalization** (Ba et al., 2016) is implemented in the same position. We report the performance averaged over five runs.

## 5 Results

### 5.1 Performance

Table 1: The performance of TextCNN$_{base}$ classifiers and the standard deviation with GraVeR applied to various pretrained word vectors.

|  | **DBpedia** | **Yah(Up)** | **Yah(Low)** | **AGNews** | **Yelp** | **IMDB** |
|---|---|---|---|---|---|---|
| Rand | 98.01±.03 | 67.41±.07 | 44.69±.15 | 88.87±.01 | 61.75±.33 | 86.42±.35 |
| + GraVeR | +0.51(.02) | +5.29(.17) | +4.76(.07) | +2.38(.39) | +1.34(.25) | +2.62(.27) |
| w/ w2v | 98.20±.22 | 67.63±.31 | 43.18±.74 | 90.28±.84 | 62.26±.16 | 86.53±.36 |
| + GraVeR | +0.26(.02) | +4.92(.20) | +6.25(.75) | +0.99(.14) | +0.69(.47) | +0.43(.37) |
| w/ glv | 98.63±.04 | 72.97±.57 | 48.46±.41 | 91.88±.11 | 63.04±.18 | 88.56±.76 |
| + GraVeR | +0.01(.03) | +0.18(.33) | +3.00(.52) | 0.00(.11) | -0.03(.10) | +0.50(.58) |
| w/ ftt | 97.80±.32 | 63.13±.54 | 38.62±.38 | 89.07±1.56 | 61.78±.18 | 75.98±3.32 |
| + GraVeR | +0.69(.02) | +9.01(.32) | +10.31(.87) | +2.21(.10) | +0.91(.31) | +9.71(.53) |

Firstly, we present the effect of GraVeR on major pretrained word embeddings, as presented in Table 1. We use three pretrained word embeddings: word2vec (w2v) (Mikolov et al., 2013a) GoogleNews-vectors-negative300.bin, GloVe (glv) (Pennington et al., 2014) glove.42B.300d.txt, fastText (ftt) (Bojanowski et al., 2016) wiki-news-300d-1M-subword.vec, and 1 random embedding.

GraVeR improves the model performance on most of the datasets, and the largest performance gain is in random embeddings. Random embeddings with GraVeR even perform better than pretrained embeddings in a few datasets. The result implies that we can learn better word representations through GraVeR in any embedding space since we train a model from scratch except for word embeddings. However, with GloVe, GraVeR is not effective on the model performance because the

Table 2: The performance of TextCNN$_{base}$ classifiers and the standard deviation with different regularization techniques. We can observe that GraVeR positively matches with the classifier and with other regularization techniques. Bold indicates the largest performance gain.

| | DBpedia | Yah(Up) | Yah(Low) | AGNews | Yelp | IMDB |
|---|---|---|---|---|---|---|
| TextCNN$_{base}$ | 98.01±.03 | 67.41±.07 | 44.69±.15 | 88.87±.01 | 61.75±.33 | 86.42±.35 |
| + WordDrop | +0.05(.03) | +0.16(.84) | -0.40(.35) | +0.33(.34) | +0.51(.16) | -0.40(.11) |
| + DO (p=0.1) | -0.01(.08) | -0.41(.44) | +0.10(.42) | +0.06(.33) | +0.43(.12) | -0.03(.34) |
| + BN | -0.16(.02) | +0.27(.31) | +0.03(.29) | -0.80(.30) | +0.49(.10) | -0.89(.45) |
| + LN | +0.17(.04) | -0.31(.25) | +0.47(.68) | -0.23(.08) | +1.14(.29) | -1.23(.38) |
| + GraVeR | **+0.51(.02)** | **+5.29(.17)** | **+4.76(.07)** | **+2.38(.39)** | **+1.34(.25)** | **+2.62(.27)** |

| | DBpedia | Yah(Up) | Yah(Low) | AGNews | Yelp | IMDB |
|---|---|---|---|---|---|---|
| TextCNN$_{base}$ | 98.01±.03 | 67.41±.07 | 44.69±.15 | 88.87±.01 | 61.75±.33 | 86.42±.35 |
| + GraVeR | +0.51(.02) | +5.29(.17) | +4.76(.07) | +2.38(.39) | **+1.34(.25)** | +2.62(.27) |
| + DO (p=0.1) | 98.00±.08 | 67.01±.44 | 44.79±.42 | 88.93±.33 | 62.18±.12 | 86.39±.34 |
| + DO&GraVeR | +0.48(.03) | +5.33(.21) | +5.11(.17) | +2.44(.13) | +1.14(.15) | +2.55(.09) |
| + BN | 97.86±.02 | 67.68±.31 | 44.72±.29 | 88.07±.30 | 62.24±.10 | 85.53±.45 |
| + BN&GraVeR | **+0.66(.01)** | +5.00(.42) | +5.17(.07) | **+2.71(.49)** | +1.29(.27) | +2.18(.11) |
| + LN | 98.18±.04 | 67.10±.25 | 45.15±.68 | 88.64±.08 | 62.89±.29 | 85.18±.38 |
| + LN&GraVeR | +0.43(.03) | **+6.50(.16)** | **+6.51(.16)** | +2.39(.10) | +0.80(.10) | **+2.99(.51)** |
| + DO&BN | 97.91±.03 | 67.87±.04 | 45.10±.29 | 88.32±.27 | 62.51±.11 | 84.89±.36 |
| + DO&BN&GraVeR | +0.58(.01) | +4.49(.12) | +4.53(.06) | **+2.71(.12)** | +0.75(.21) | +2.71(.46) |
| + DO&LN(TextCNN$_{tune}$) | 98.49±.06 | 69.26±.31 | 46.85±.18 | 90.06±.35 | 63.24±.34 | 86.11±1.17 |
| + DO&LN&GraVeR | +0.24(.07) | +5.83(.26) | +5.95(.38) | +1.33(.25) | +0.70(.17) | +2.67(.36) |

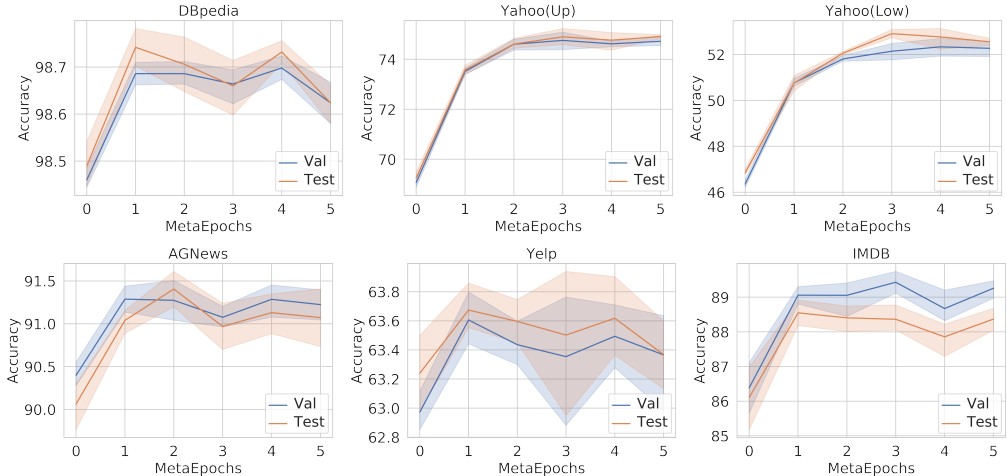

Figure 2: Training curves in the classification datasets. As GraVeR makes the model fit to the validation set, the performance on the test set increases.

distribution of word vectors in the pretrained embeddings is already good enough for the tasks. In contrast, GraVeR makes substantial improvements when using relatively poor embeddings for the tasks.

The comparison with regularization techniques is presented in Table 2 (Top). The result shows that the performance gain by GraVeR is larger than other regularization methods. We also present the results when our method is combined with other regularization methods in Table 2 (Bottom). The results show that GraVeR positively matches with the other regularization techniques, further improving the model performance. Figure 2 also reports that our method definitely increases validation performance, which results in improvements in the test set.

Comparisons of a recent model (e.g., BERT) without extra data resources and word embedding perturbation methods will be discussed in Further Analysis (§6).

Table 3: List of top-20 nearest words of a cue word (`love`) in initial embedding (`Initial`), after fine-tuned once (`FineTuned`), and our method `GraVeR` in DBpedia dataset. The differences between initial embedding and the other methods are marked in underlined. The differences between fine-tuned once and GraVeR are marked in bold. GraVeR further changes the distribution of word vectors.

| Method | Top-20 Nearest Words with `love`(Cosine Similarity) |
|---|---|
| `Initial` | loved(.77),i(.73),loves(.73),know(.73),loving(.73),really(.72),always(.72),want(.72), hope(.71),think(.71),like(.7),me(.7),'m(.69),never(.69), hate(.69),happy(.69),so(.69),n't(.69),just(.69),wish(.69) |
| `Fine Tune` | know(.72),loved(.72),really(.69),you(.68),always(.68),loving(.66),wish(.66),i(.65),say(.64), hate(.64),loves(.64),me(.64),my(.64),everything(.64),believe(.63),like(.63),adore(.63), so(.63),remember(.63),want(.63) |
| `GraVeR` | know(.7),loved(.68),you(.64),wish(.64),remember(.63),say(.61),**think**(.61), loving(.61),me(.6),want(.6),loves(.6),i(.6),believe(.6),**fun**(.6),hate(.6), **wanted**(.6),really(.59),always(.59),my(.59),**hope**(.59), |

## 5.2 Word Distributions

To analyze GraVeR with respect to word representation, we extract the word representations updated on DBpedia dataset and present the list of top-20 nearest words of a cue word in Table 3. In order to reduce the effect of randomness, we use GloVe for initial embedding. We can see that the word vectors are further fine-tuned and even find other similar words not shown in the embedding fine-tuned once. These results imply that our method can change the word vector distribution to be further fine-tuned to the tasks. We also present another cue word results and visualization of top-100 nearest word vector distribution using t-SNE (Maaten & Hinton, 2008) in Appendix.

## 6 Further Analysis

From this Section, we mainly use TextCNN$_{tune}$ to show the effect of GraVeR on the model whose performance is similar to state-of-the-art models.

### 6.1 Random Noise and Bias

Table 4: The performance of TextCNN$_{tune}$ and the standard deviation with GraVeR according to the random noises and bias. ▷ indicates our default setting `noise scale 1`, which means the random values sampled from uniform distribution the range between -1 and 1.

| | DBpedia | Yah(Up) | Yah(Low) | AGNews | Yelp | IMDB |
|---|---|---|---|---|---|---|
| Only Re-train. | 98.47±.05 | 68.73±.35 | 45.55±.42 | 89.85±.22 | 63.21±.20 | 85.43±1.04 |
| Gaussian Noise | 98.71±.03 | 74.12±.54 | 51.27±.34 | 90.91±.31 | 63.64±.08 | 88.55±.19 |
| No Additive Bias | 98.65±.02 | 73.41±.10 | 49.57±.81 | 90.73±.38 | 63.63±.12 | 88.54±.27 |
| NoiseScale 0. | **98.73±.01** | 74.78±.10 | 52.17±.49 | 90.93±.21 | 63.83±.16 | 88.55±.33 |
| ▷ NoiseScale 1. | **98.73±.07** | **75.09±.26** | **52.80±.38** | **91.40±.25** | **63.94±.17** | **88.77±.36** |
| NoiseScale 5. | 98.72±.03 | 74.30±.34 | 51.27±.25 | 90.72±.52 | 63.34±.27 | 88.10±.49 |
| Emb Perturbation | 98.46±.03 | 68.93±.33 | 45.99±.23 | 89.39±.38 | 63.03±.11 | 86.27±.53 |

**Noises.** The range of random values filled in the maskers also affects the performance. We first re-train the model without any modification to observe the effect of the re-training processes. The model shows slightly better performance within a few meta-epochs, but it becomes overfitting, as we expected (see Table 4). The performance when we use Gaussian noise instead of the uniform distribution is also presented. It shows a comparable but slightly worse result than our default settings. When the noise range is 0, the noise only consists of the additive bias, which also shows marginally worse performance. The word perturbation method performs slightly better than just re-training, but there are large gaps between variations of GraVeR.

## 6.2 HYPERPARAMETERS

Table 5: The performance of TextCNN and the standard deviation with GraVeR according to the masking rate, which determines how much the random maskers move to the next frequently used words. ▷ indicates our default setting that masks 20% of vocabulary at once.

|  | DBpedia | Yah(Up) | Yah(Low) | AGNews | Yelp | IMDB |
|---|---|---|---|---|---|---|
| MaskRate 0.1 | **98.78±.06** | 74.86±.20 | 52.45±.21 | 91.06±.20 | **64.06±.22** | 88.50±.34 |
| ▷MaskRate 0.2 | 98.73±.07 | **75.09±.26** | **52.80±.38** | **91.40±.25** | 63.94±.17 | **88.77±.36** |
| MaskRate 0.5 | 98.60±.02 | 73.06±.33 | 49.87±.42 | 91.02±.28 | 63.60±.14 | 88.44±.20 |
| MaskRate 1.0 | 98.65±.02 | 70.89±.28 | 47.17±1.31 | 90.99±.18 | 63.75±.02 | 87.83±.34 |

**Masking Rate**   The amount of noise added by GraVeR is an important factor in that some noises should be small enough to be corrected during the re-training processes, while other noises should be large enough to change the word vector distribution. We first change the masking rate of how much random maskers move in every re-training process. The larger the masking rate becomes, the more words are masked in a re-training process, so the noise increases. Conversely, the amount of noise decreases as the masking rate becomes small. The effect of the masking rate is presented in Table 5. `Masking Rate=0.1` also shows good performance, but it needs 2x re-training processes more than `Masking Rate=0.2`. We thus use 0.2 as a default.

Table 6: The performance of TextCNN classifiers and the standard deviation with GraVeR according to the gradualness policy. The default method is to increase the number of maskers when the validation performance decreases.

|  | DBpedia | Yah(Up) | Yah(Low) | AGNews | Yelp | IMDB |
|---|---|---|---|---|---|---|
| Proposed | **98.73±.07** | **75.09±.26** | **52.80±.38** | **91.40±.25** | **63.94±.17** | 88.77±.36 |
| No Gradualness | 98.68±.08 | 74.17±.24 | 52.23±.56 | 91.28±.11 | 63.74±.18 | 88.47±.36 |
| Reversed Grad. | 98.69±.06 | 73.97±.40 | 52.07±.23 | 91.00±.34 | 63.84±.14 | **88.96±.47** |
| Both Grad. | 98.66±.02 | 74.23±.48 | 51.87±.49 | 91.09±.29 | 63.55±.36 | 88.79±.22 |

**Gradualness Policy**   Our proposed method is to increase the number of maskers when the validation performance decreases in order to make noises on the words masked in the previous re-training process again. Otherwise, we simply move to the next frequently used words. We try changing the gradualness that (1) do not have gradualness, (2) have gradualness only when the validation performance increase, which is reverse to our proposed method, and (3) always have gradualness regardless of the validation performance. The result is presented in Table 6. Among the ablations, our proposed approach performs the best.

## 6.3 ON OTHER MODEL ARCHITECTURE

Table 7: The performance of Transformer-based classifier and the standard deviation.

|  | DBpedia | Yah(Up) | Yah(Low) | AGNews | Yelp | IMDB |
|---|---|---|---|---|---|---|
| TransformerClf | 98.42±.03 | 76.11±.02 | 57.68±.08 | 89.54±.31 | 60.07±.25 | 84.49±.93 |
| +GraVeR | 98.50±.02 | 76.38±.24 | 57.92±.25 | 90.40±.28 | 60.26±.25 | 85.75±.85 |

We opt our method to the transformer-based classifier. The transformer classifier has 300 embedding dimension with positional-embedding, 32 batch size, 512 sequence length. It also has 10 multi-heads but only 1 encoder layer. Stacking more than 2 encoder layers harms the performance; We guess that the number of training set is not enough to train the model parameters. We do average-pooling to the encoded sentence vectors and use linear layer in order for classification. The other parameters follow the default setting of Pytorch nn.TransformerEncoderLayer. Table 7 shows that GraVeR works well even in other model architectures.

Table 8: The performance of TextCNN$_{tune}$ classifiers and the standard deviation with pretrained word vectors trained from only training data and comparison with pretrained word vector trained from external resources.

|  | DBpedia | Yah(Up) | Yah(Low) | AGNews | Yelp | IMDB |
|---|---|---|---|---|---|---|
| `GraVeR(Sp)` | 98.73±.07 | 75.09±.26 | 52.80±.38 | **91.40±.25** | **63.94±.17** | **88.77±.36** |
| `w2v (Gen)` | 98.61±.03 | 70.45±1.02 | 46.74±.66 | 91.27±.23 | 62.73±.23 | 85.07±1.54 |
| `w2v (Sp)` | +0.08(.04) | +3.72(.43) | +4.25(.51) | -0.32(.34) | +0.52(.33) | -0.02(1.18) |
| `glv (Gen)` | **98.79±.07** | 75.77±.41 | 52.83±.57 | **91.37±1.10** | 63.59±.32 | 87.67±.60 |
| `glv (Sp)` | -0.54(.34) | -2.47(1.01) | -4.72(1.84) | -2.41(.77) | -0.59(.43) | -5.19(1.09) |
| `ftt (Gen)` | 98.43±.13 | 64.56±.31 | 40.91±.44 | 89.43±.82 | 62.75±.11 | 79.16±1.02 |
| `ftt (Sp)` | +0.17(.11) | +9.46(.56) | +9.52(.58) | +1.49(.14) | +0.41(.23) | +4.57(1.53) |
| `BERT (Gen)` | 99.20±.01 | 81.75±.41 | 65.83±.37 | 93.58±.23 | 69.25±.15 | 93.24±.35 |
| `BERT (Sp)` | 98.57±.13 | **77.19±.32** | **58.11±.58** | 90.35±.21 | 60.99±.22 | 87.17±.34 |

### 6.4 PRETRAINING BY TRAINING DATA.

Table 8 shows the classification performance according to the usage of pretrained word embeddings. `Gen` means general representation trained on a large corpus, which is the pretrained embeddings what we mentioned in §5.1 and `bert-base-uncased` from huggingface (Wolf et al., 2019). `Sp` denotes specialized representation trained using only training data. We set the hyperparameter for training as default settings used in their API.

Despite a small number of parameters, TextCNN$_{tune}$ with GraVeR shows comparable performance with BERT (`Sp`). GraVeR's improvement is even similar to glv (`Gen`), which means GraVeR can complement the information from external resources, in this case, Common Crawl (42B tokens, 1.9M vocab). Although there is a performance gap with pretrained BERT, the advantages of GraVeR are (1) do not need any extra data resources, and (2) do not need to use very deep model architectures consisting of a bunch of parameters. Therefore, GraVeR must be an attractive option when using pretrained word embeddings.

## 7 DISCUSSION

**Re-training Cost.** Although GraVeR shows strong performance, its re-training processes take `1/[MaskingRate]` more times than conventional training process. However, we showed that a small model with GraVeR even performs on par with recent huge models. When considering the parameter size, for example, TextCNN$_{tune}$ has 136K parameters and it needs five times re-training process in our default setting `MaskingRate=0.2` while BERT has 110M parameters with one fine-tuning process. Then, the parameters need to be trained are 780K and 110M, respectively; i.e. 141 times cheaper. Furthermore, the training time for small model must be faster since it can be trained with larger mini-batch size.

**As a representation learning.** GraVeR's representations are learned from a given training set only. In Table 8, general word embeddings trained from large data shows worse performance than domain (or data) specialized word embeddings. That is, in order to solve the task, the word representation should be specialized in given data. By using GraVeR, we can easily get specialized (further fine-tuned) representation only with the random noise including bias and the iterative training processes. Besides, we believe that using a sophisticated noising trick instead of simple random makes further improvement in GraVeR.

## 8 CONCLUSION

We propose GraVeR, which adds random noises and bias to word embeddings in order to change the word vector distribution and regularize a model. Through the re-training process, we can make the use of noises to learn better representations. In the experiments, as the model incrementally fits the validation set, GraVeR largely improves model performances. We expect that our general training approach can be used to a various models to improve model performances.

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
