# OpenReview forum: "Ruminating Word Representations with Random Noise Masking"
_ICLR.cc/2021/Conference — Reject_

### Official Review · AnonReviewer3 · 2020-10-20
**Unclear exposition**

**Rating:** 3
**Confidence:** 4

**Review:**


##########################################################################

Summary:

This paper presents a simple meta-learning method for iteratively fine-tuning
word embeddings. The method works by adding noise and bias terms to the learned
embeddings after each training session, and initializing the same model on the
same task with the resulting representations for further fine-tuning. Noise is
added to a subset of the vocabulary, while bias is added to every element of it.
The process is repeated until noise has been added to all the elements of the
vocabulary.

Results show that the proposed method consistently improves results in the set
of classification tasks in which it was tested.


##########################################################################

Reasons for score:

However, the paper is not very thorough in its exposition and leaves too many
questions without answer. In all, it is difficult to assess the real
contribution of the suggested method.


##########################################################################

Pros:

- Method could be useful to the community working on non-contextualized word
  representations.

##########################################################################

Cons:

- The proposed method is unclear.
- Analyses are lacking

#########################################################################

Comments, suggestions and questions for the authors:

- p. 1, sec. 1, par. 1: I do not understand the difference between
  "starting with better representations" and "building more sophisticated
  architectures". For the former you cite ELMo as an example, but ELMo was both
  a new architecture and training methodology, hence my confusion with
  qualifying it as "starting with better representations". For the latter you
  cite the Transformers, which makes more sense as a new architecture. But then,
  how would you classify BERT, being both a new architecture sharing transformer
  components, and a new training methodology? (This is more of a rhetorical
  question; I suggest clarifying the points above directly in the paper.)

- p. 1, sec. 1, par. 4: Did you take into consideration phenomena such as
  catastrophic forgetting (https://arxiv.org/abs/1312.6211), when re-training?
  Even though you are not changing the training task, I think it would be
  valuable to at least mention this.

- p. 2, sec. 2.1, par. 1: Mikolov et al. 2013b make no mention of the
  distributional hypothesis, and even if they did, there is no implication
  between this hypothesis and the fact that pretrained embedding vectors are
  composed of the pairs you mention.

- p. 2, sec. 2.1, par. 2: A reader not deeply familiarized with the differences
  between contextualized word representations, and pre-trained word-vectors will
  probably have a hard time understanding this paragraph. I suggest making it
  clear that contextualized representations in the context of ELMo and BERT are
  actually pre-trained models, and not only pre-trained word representations
  such as GloVe, word2vec, and fasttext.

- p. 2, sec. 2.2, par. 2: References to previous works doing "Word Dropping"
  would be a welcome addition to your paper. Is this paragraph an allusion to
  stopword removal, or something else?

- p. 3, sec. 3.2, par. 2: `MaskingRate` is a hyperparameter of the noise
  maskers, but in sec. 3.1, par. 2 it says that the bias is weighted by
  1/[`MaskingRate`]. I suggest clarifying the differences between noises and
  biases, and their relationship to maskers.

- p. 3, sec. 3.2, par. 3: You mention you add a bias to the fine-tuned word
  embeddings, in order to take them to the same embedding space. What do you
  mean exactly with this? Addition does not change the dimensionality of the
  summands, therefore adding something to W' will only shift its values and not
  modify their embedding space.

- p. 5, sec. 4.2, par. 3: How do you initialize word embeddings randomly? Is it
  also by sampling from the uniform distribution between -1 and 1?

- p. 5, sec. 5.1, par. 2: It is unclear what you mean with "since we train a
  model from scratch except for word embeddings."

- p. 6, sec. 5.1, par. 2: You mention that GraVeR makes substantial improvements
  when using relatively poor embeddings, but I can see that GraVeR has a
  significant impact on fasttext embeddings (Table 1), which I would not call
  poor.

- p. 6, Table 2: I suggest specifying what the values between parentheses and
  after the $\pm$ are; standard deviation? confidence interval? if so, with
  which confidence level?. Same suggestion for Table 3.

- p. 7, sec 5.2: Why did you decide to show cherry-picked results for GloVe,
  despite your method not working well with these pre-trained embeddings, as
  shown in Table 1? Why not show results for fasttext embeddings instead?

- p. 7, Table 4: If NoiseScale 1 means that the values from the masker were
  sampled from Uniform(-1; 1), as explained in section 3.2, I assume that
  NoiseScale 5 implies Uniform(-5; 5). If this is true then what does NoiseScale
  0 imply?

- p. 7, sec. 6.1, par. 1: How can you tell from Table 4 that the model
  overfits?

- p. 8, Table 5: As mentioned previously, I am confused by the choice of
  parenthesis versus plus-minus symbols, and the meaning of the numbers they
  refer to. Why do you use $\pm$ for every Gen variant, but parentheses for Sp?
  why do you use $\pm$ for BERT (Sp), despite the previous? Also, why did you
  decide to invert the Gen-Sp row order in the BERT entry?

- I am a bit confused about the role Maskers play in your model. You mention
  them for the first time in the caption of Figure 1, without much context. Then
  you mention that you "add maskers filled with random values to a portion of
  W'", which makes me think that maskers are nothing other than random vectors,
  but I am not 100% sure. The paper goes on assuming that the reader has a clear
  understanding of what maskers are, but I personally found this not to be the
  case. Further, maskers are not mentioned in Algorithm 1, only `MaskedWords`
  and `NextFrequentlyUsedWords`, which I guess are somehow related to maskers,
  but again, I can't be sure exactly how just by reading the paper.

- Another pain point related to the above is that you superficially mention (in
  Algorithm 1, and the two paragraphs before it), that word frequency somehow
  conditions which words are masked, but no details are given as to how this
  happens.

- I found the tables to be a bit overcrowded and difficult to parse in general.
  I suggest trying to simplify them to better get your point across.

- I think the Table 1 mentioned in the Appendix should be Table 6.


#########################################################################

Some typos:

- Abstract: "some of noises" -> "some noises"
- Abstract: What do you mean with the last sentence?
- p. 1, sec. 1, par. 2: "the general and meaning of words" -> "the general
  meaning"
- p. 1, sec. 1, par. 4: "to the next re-training process" -> "in the next
  re-training process"
- p. 1, sec. 1, par. 4: What do you mean with "the model must be overfitted"?
- p. 2, sec. 2.2, par. 3: "It can regularize the models,[...]re-usability." is
  not a grammatically correct sentence.
- p. 2, sec. 2.3, par. 2: In "Also, dropout discourses the weights" I think you
  meant to write something other than "discourses".
- p. 3, sec. 3.1, par. 2: "best-performed" -> "best-performing"
- p. 4, sec. 3.2, par. 4: I think the sentence "Although this approach cannot
  warrant moderate noise, we take a safe approach." is not clear enough.
- p. 4, sec. 4.1, par. 1: "in Appendix" -> "in the Appendix"
- p. 5, sec. 4.2, par. 1: "with the 32 channels" -> "with 32 channels"
- p. 5, sec. 4.2, par. 1: "max-pooled" -> "max-pooling"
- p. 5, sec. 4.2, par. 2: "small parameters" -> "few parameters"
- p. 5, Table 1 caption: "vector" -> "vectors"

---

### Official Review · AnonReviewer2 · 2020-10-29
**A novel regularization for embeddings in CNN-based text classification.**

**Rating:** 4
**Confidence:** 4

**Review:**

It's already known that embeddings like word2vec and glove are biased [1] and needs postprocessing for better performance. This paper designed a novel approach to do embedding normalizations. After each round of training, noise is intentionally introduced to perturbate the finetuned parameters. Afterwards, another round of training starting from perturbated local optima could in potential converge to a better one. This method is validated via CNN-based text classification.

Pros:
+ The idea to iterate the training process is interesting and the proposed approach shows reasonable consistent improvement in different domains, comparing to existing regularization methods.

Cons:
- Text classification is a standard NLP task and CNN is not the only solution. The performance boost on top of RNN or transformer-based approach could make the regularization more convincing.
- The power of word2vec and glove comes from the fact that they're trained using a large unsupervised corpus. The regularization method proposed in this paper, however, requires supervised labels on the training and validation set. The proposed approach is more stronger if the idea can be further generalized to pretraining a better set of word embeddings.
- Retraining cost. Even though the model size of CNN is smaller than BERT so the cost on retraining seems to be reasonable, there is still a performance gap between multi-round and one-shot BERT.

Minor comment: the notation is not clear in the paper. In the algorithm, one perturbation is added to W' but the other one is added to W, is this a typo?

[1] Mu, Jiaqi, Suma Bhat, and Pramod Viswanath. "All-but-the-top: Simple and effective postprocessing for word representations." arXiv preprint arXiv:1702.01417 (2017).

---

### Official Review · AnonReviewer1 · 2020-10-29
**Ok paper, but not good enough**

**Rating:** 4
**Confidence:** 4

**Review:**

This paper proposes to improve word embedding by iteratively 1) adding noises to the trained word embeddings, and 2) retraining the model with the noised embeddings.

While the paper has some interesting points, I feel the paper lacks novelty because adding noises as regularization is a well-known, and the iterative incurs significant costs. Besides, I found the experiment comparison unconvincing. Thus, I give this paper the overall rating "rejection".

Comment:
1) In the introduction, the author claims the embeddings will often stuck in local minimums during training as one motivation for this work. It would be better to have some experimental demonstration of this claim.
2) The iterative approach requires an task-specific validation set, and consults the validation accuracy multiple times during the whole training process. This provide an unfair advantage for the proposed method, compared to the baselines which does not involves any task-specific signals.

---

### Decision · Program_Chairs · 2021-01-07
**Final Decision**

**Decision:**

Reject

**Comment:**

This paper proposes adding noise regularization, iteratively during training to word embeddings.
The method is evaluated on CNN-based text classification.
Overall, there is novelty in the proposed method, however there are concerns about the experiments and analysis of the proposed approach.